# Patterns of smallpox mortality in London, England, over three centuries

**Olga Krylova**[ID][1][¤], **David J. D. Earn**[ID][1,2]*

**1** Department of Mathematics and Statistics, McMaster University, Hamilton, Ontario, Canada, **2** M.G. DeGroote Institute for Infectious Disease Research, McMaster University, Hamilton, Ontario, Canada

¤ Current address: Advanced Analytics Branch, Canadian Institute for Health Information, Ottawa, Ontario, Canada

* earn@math.mcmaster.ca

**Data Availability Statement:** All data are available at https://github.com/davidearn/London_smallpox and at the International Infectious Disease Data Archive at http://iidda.mcmaster.ca.

**Funding:** Natural Sciences and Engineering Research Council of Canada (NSERC, http://www.

## Abstract

Smallpox is unique among infectious diseases in the degree to which it devastated human populations, its long history of control interventions, and the fact that it has been successfully eradicated. Mortality from smallpox in London, England was carefully documented, weekly, for nearly 300 years, providing a rare and valuable source for the study of ecology and evolution of infectious disease. We describe and analyze smallpox mortality in London from 1664 to 1930. We digitized the weekly records published in the London Bills of Mortality (LBoM) and the Registrar General's Weekly Returns (RGWRs). We annotated the resulting time series with a sequence of historical events that might have influenced smallpox dynamics in London. We present a spectral analysis that reveals how periodicities in reported smallpox mortality changed over decades and centuries; many of these changes in epidemic patterns are correlated with changes in control interventions and public health policies. We also examine how the seasonality of reported smallpox mortality changed from the 17th to 20th centuries in London.

## Introduction

Smallpox was declared eradicated 40 years ago, in 1980 [1], after unparalleled devastation of human populations for many centuries [2,3]. Until the 19th century, smallpox is thought to have accounted for more deaths than any other single infectious disease, even plague and cholera [2–7]. In the city of London, England alone, more than 320,000 people are recorded to have died from smallpox since 1664.

Investigation of smallpox dynamics in the past is important not only for understanding the epidemiology of a disease that has been exceptionally important in human history but also in the context of the potential for its use as a bioterrorist agent in the future [8–12]. From the historical perspective, much can be learned by relating patterns of smallpox outbreaks to demographic changes and uptake of preventative measures [13–16] and to wars and other historical events [4]. More broadly, appreciation for how intentional interventions and coincident events might impact disease dynamics is important when attempting to control the spread of any infection, including new diseases such as Coronavirus Disease 2019 (COVID-19) [17–19].

nserc-crsng.gc.ca). NSERC Discovery Grant to DE.
NSERC Postgraduate Scholarship to OK. J. S.
McDonnell Foundation (JSMF, http://jsmf.org)
Research Grant to DE. The funders had no role in
study design, data collection and analysis, decision
to publish, or preparation of the manuscript.

**Competing interests:** The authors have declared
that no competing interests exist.

**Abbreviations:** ACM, all-cause mortality; CFP, case
fatality proportion; COVID-19, Coronavirus Disease
2019; EMD, Empirical Mode Decomposition;
IIDDA, International Infectious Disease Data
Archive; IMF, intrinsic mode function; LBoM,
London Bills of Mortality; RGWR, Registrar
General's Weekly Return.

Previous work on smallpox dynamics in London has either been based on annual records [20–22] or been restricted to a few decades [23–25]. We present and examine 267 years of weekly records of smallpox mortality in London, beginning in 1664. The data span an early era before any public health practices were in place, the introduction of variolation and then vaccination, and then the decline of smallpox mortality until it became an extremely unusual cause of death. The temporal resolution in the data enables analysis of short-term fluctuations and seasonality, which are smoothed out by yearly data.

Our statistical descriptions of the weekly smallpox data will help sharpen and quantify research questions concerning the mechanistic origin of changes in the temporal patterns of epidemics [13–16]. In addition, we present a timeline of major historical events that occurred during the epoch we have studied. Overlaying the historical timeline with smallpox mortality and prevention patterns provides an illuminating view of three centuries of smallpox history.

## Smallpox background

Smallpox is an acute, highly contagious, and frequently fatal disease. The name "small-pox" was first used in England at the end of the 15th century to distinguish it from syphilis, which was known as "great-pox" ([2], pp. 22–29). The disease is caused by two variants of the *Variola* virus, which differ substantially in severity of symptoms and case fatality proportion (CFP) ([1], pp. 1–68; [26], pp. 525–527). *Variola major*, which was the only known smallpox type until the beginning of the 20th century, had a CFP of 5% to 25% and occasionally higher ([1], p. 4). *Variola minor* (also known as *alastrim*), which was first recognized in 1904 [2], was less virulent and had a CFP of about 1% or less. *Variola minor* was the only endemic type of smallpox present in England after 1920 ([2], pp. 8, 97; [1], pp. 243). By 1935, transmission of smallpox within England appeared to have ended; further naturally acquired infections are believed to have arisen only from importation [2].

Among those who survived it, morbidity from smallpox was severe in many cases; victims could be left blind or disfigured for life [27]. In the pre-vaccination era, there were attempts to reduce morbidity and mortality from smallpox by a method initially known as **inoculation**, and later given the more specialized term **variolation**. This procedure involved deliberately infecting a healthy individual with smallpox virus taken from a pustule or dried scabs of a person suffering from smallpox [3].

Edward Jenner's discovery of a smallpox vaccine in 1796 [28] was a major milestone, not only for smallpox control but also for modern medicine more generally, as it inspired the development of vaccines for many other pathogens. Jenner's vaccine provided a safer, cheaper, and more effective alternative to variolation. His original method, which he called "vaccine inoculation" [28], was to inject a person with cowpox virus (*Vaccinia*, from the Latin *vacca* for cow).

The existence of a smallpox vaccine was the key factor that made eradication of the disease an achievable goal. Other important factors included the absence of asymptomatic infections, easy recognition of the disease from its symptoms, absence of an animal reservoir, and relatively low infectivity [29]. The World Health Organization launched its eradication campaign in 1967 [1,30]. Ten years later, the world's last endemic smallpox case was registered in Somalia. In 1980, a Global Commission declared smallpox eradicated [31]. This was the first disease to be eradicated entirely by human efforts. The only remaining viral samples were stored in laboratories in Russia and the United States [11].

## Data

Registration of burials in England began in 1538, when Thomas Cromwell introduced parish registers [32], but systematic summaries of deaths categorized by cause were not published until later, and only for a few towns, in **Bills of Mortality**. The bills reported Anglican burials,

not all deaths. Later, with the creation of the office of the Registrar General in the mid-19th century, a formal system of registration of all deaths was developed and maintained.

We accessed original documents in London, England, in the Guildhall Library, the British Library, the Wellcome Library, and the London Metropolitan Archive. We digitized weekly reported birth and death records for London throughout the period over which smallpox was listed as a cause of death (1661 to 1934). The last smallpox death reported in London was in the week beginning 17 February 1934. The last year when more than one smallpox death was reported in a single week was 1930, so we do not present data or analyses after 1930 (in total, only 7 smallpox deaths were reported from 1931 to 1934).

Until 1752, the New Year was celebrated on 25 March in England [33]. However, we always indicate years following the modern practice of defining New Year's Day to be 1 January. All data analyzed in this paper are shown in Fig 3 and are available at the International Infectious Disease Data Archive (IIDDA; http://iidda.mcmaster.ca) and at https://github.com/davidearn/London_smallpox.

## London Bills of Mortality

The London Bills of Mortality (LBoM) included information about baptisms and church burials, categorized by cause of death (Fig 1). The Company of Parish Clerks published the bills, based on counts collected from the individual Anglican parish registers [34,35]. Weekly bills began to be published frequently in 1604 [36], but not without long gaps until 1661. A nearly continuous weekly sequence of bills survives starting 18 October 1664.

The accuracy of the LBoM has been considered by a number of historians and demographers [23,37–42]. Shortcomings that are often highlighted include:

**Omissions**. Only burials within Anglican grounds were included; Roman Catholics, Jews, Quakers, and other nonconformists were excluded [39–42].

**Geographical coverage**. The LBoM did not include some fast-growing parishes and did not reflect expansion of London's boundaries due to population growth [40,42].

**Accuracy of parish returns**. Parish returns were sometimes missing or inaccurate [40,42,43].

**Exported burials**. Some Londoners were buried outside the city boundaries [23,40,41].

**Declining accuracy**. After 1790, an increasing number of poor Londoners were buried in non-Anglican grounds and were not registered [39].

**Unsubmitted returns**. Beginning around 1830, some parishes stopped submitting their returns [43]. (In Fig 3, we annotate the period from 1790 to 1841 as a "Progressive collapse of the parish registration system".)

All of these data quality issues led to underreporting of mortality. Nevertheless, the data from the LBoM are "probably more complete and more accurate than any available elsewhere in England in that time" [37] and remain the most comprehensive summaries of baptisms and burials of the 17th and 18th centuries in London.

While the bills are certainly not a perfect record of mortality, it is reasonable to assume—as we do in this paper—that the temporal pattern of smallpox burials quantified in the LBoM, is roughly proportional to the true historical smallpox mortality in London. We implicitly assume that changes in the degree of underreporting of smallpox deaths were sufficiently slow that we can make inferences about seasonality of smallpox deaths and the frequency of epidemics. However, from the LBoM, we can estimate only a lower bound for the total burden of smallpox deaths in London between 1661 and 1841.

## The Diseases and Casualties this Week.

| | |
|---|---|
| ABortive | 5 |
| Aged | 43 |
| Ague | 2 |
| Apoplexie | 1 |
| Bleeding | 2 |
| Burnt in his Bed by a Candle at St. Giles Cripplegate | 1 |
| Canker | 1 |
| Childbed | 42 |
| Chrisomes | 18 |
| Consumption | 134 |
| Convulsion | 64 |
| Cough | 2 |
| Dropsie | 33 |
| Feaver | 309 |
| Flox and Small-pox | 5 |
| Frighted | 3 |
| Gowt | 1 |
| Grief | 3 |
| Griping in the Guts | 51 |
| Jaundies | 5 |
| Imposthume | 11 |
| Infants | 16 |
| Killed by a fall from the Belfrey at Alhallows the Great | 1 |
| Kingsevil | 2 |
| Lethargy | 1 |
| Palsie | 1 |
| Plague | 7165 |
| Rickets | 17 |
| Rising of the Lights | 11 |
| Scowring | 5 |
| Scurvy | 2 |
| Spleen | 1 |
| Spotted Feaver | 101 |
| Stilborn | 17 |
| Stone | 2 |
| Stopping of the stomach | 9 |
| Strangury | 1 |
| Suddenly | 1 |
| Surfeit | 49 |
| Teeth | 121 |
| Thrush | 5 |
| Timpany | 1 |
| Tissick | 11 |
| Vomiting | 3 |
| Winde | 3 |
| Wormes | 15 |

Christned { Males — 95, Females — 81, In all — 176 }
Buried { Males — 4095, Females — 4202, In all — 8297 } Plague — 7165

Increased in the Burials this Week — 607
Parishes clear of the Plague — 4    Parishes Infected — 126

The Assize of Bread set forth by Order of the Lord Maior and Court of Aldermen, A penny Wheaten Loaf to contain Nine Ounces and a half, and three half-penny White Loaves the like weight.

**Fig 1. Example of one of the London Bills of Mortality.** Burials by cause for the week ending 26 September 1665, during the Great Plague of London. Five deaths from "Flox and Small-pox" are listed. *Photo courtesy of the Public Domain Review*, https://publicdomainreview.org/collection/londons-dreadful-visitation-bills-of-mortality.

### Reporting of smallpox deaths

Burials are tabulated by cause in the LBoM, but the Bills predated formal classifications of disease, casting doubt on the reliability of the listed causes [37,40,44]. However, smallpox records are likely among the most accurate in the LBoM, due to the unique and easily identifiable presentation of the disease ([36], p. 530; [37,40,43]).

Before 1701, smallpox was listed in the weekly LBoM under the heading "flox and small-pox". "Flox" is an older term that referred to a rare type of smallpox infection involving especially severe symptoms, including hemorrhaging, and high case fatality (approximately 96%) ([36], p. 436). After 1701, "smallpox" was used consistently. Other disease names that occur in the bills and are considered by some to be associated with smallpox are "flux" and "bloody flux". Razzell [3] suggested that bloody flux was a name used for hemorrhagic smallpox and that it was considered a distinct disease ([3], p.104). However, Creighton [36] described bloody flux as an old name for dysentery and not as something related to smallpox ([36], p.774). Other historians also refer to "bloody flux" and "flux" as diseases not related to smallpox, but rather the names for dysentery and diarrhea respectively ([45], p. 401; [46], p. 45; [47], p. 611). In any case, mortality from "bloody flux" and "flux" was negligible compared to smallpox (<5,000 deaths in total from "bloody flux" and "flux" compared to 322,219 total smallpox deaths).

Consequently, even if they were truly smallpox deaths, including them would not significantly influence our findings. We therefore used the sum of "flox and smallpox" and "smallpox" records and did not include "bloody flux" and "flux" in our data.

### Transition to the Registrar General's Weekly Returns

By the end of the 18th century, it was evident that a better system of collection of vital statistics was needed in England. The accuracy of the old system of parish records had been compromised by the rapid growth of London's population [48]. The Registrar General's Office was created in 1836 by the Births and Deaths Registration Act [49] to provide a comprehensive and accurate national registration system of births, marriages, and deaths, including causes of death ([50], p. 77).

The new (national) system of civil registration began in 1837, and the first Registrar General's Weekly Return (RGWR) was published for the week ending 11 January 1840 ([51], p. 119). Unlike the LBoM, the new registration system included all deaths (not only burials), all sectors of the population (not only Anglicans), and greater geographical coverage. Initially, five additional parishes were included [39]. Other areas were added when the RGWRs were established, and this expanded area was used to define the metropolis in the 1841 census [52]. A thorough review of the quality of data in the Registrar General's Returns is provided by Hardy [44].

In 1841, 234 smallpox deaths were reported in the LBoM. In 1842, only 34 smallpox deaths were reported in the LBoM, whereas 357 were reported in the RGWR. We use the RGWR from January 1842 onwards.

### Heaping during the transition period

From the middle of the 18th century, there was often a week late in the year with a remarkably high number of deaths reported in the LBoM. This phenomenon is well known ([53], p.14)

and resulted from backlogged records being submitted together (**heaped**), typically in early December in the last reporting week of the year.

To address the problem of heaping, we examined each year from 1760 to 1841 and performed the following steps:

- We considered the number of reported smallpox deaths from week 45 of the focal year to week 5 of the next year.

- We classified as a **heap week** any week from week 45 to week 5 in which reported smallpox mortality was more than three standard deviations from the median of these weeks.

- We verified by visual inspection that the automatically detected heap weeks were indeed unusual, and we looked for visually apparent heap weeks that were missed.

- We replaced the reported smallpox deaths in heap weeks with the average of the previous and following weeks.

- We calculated the difference between original heaped count and the replaced value (the **excess** due to heaping) and redistributed this number of smallpox deaths in proportion to reported smallpox throughout the year (so the adjusted counts have noninteger values). This redistribution ensured that the original and revised time series contained the same annual numbers of smallpox deaths. (We separately considered redistributing the excess uniformly throughout the year and did not detect any differences in our results.)

The first year with a heap week identified in this way was 1768, and the last was 1841. Thirty-nine years with heap weeks were discovered by the three standard deviations criterion. Another 14 years with heap weeks were identified visually, giving a total 51 years involving heaping. From 1793 to 1841, almost every year had a heap week.

## Missing weeks

The mortality bills for some weeks have been lost. Fortunately, all gaps are small (typically 1 to 5 weeks) with the largest gap being 9 weeks. We filled all gaps using linear interpolation, so there are no missing values in the final time series.

## Consistency checks

Annual summaries of mortality were published in London from 1629 onwards. Initially, these were annual Bills of Mortality, and later, they were annual summaries of the Registrar General's Returns. Creighton [36] tabulated annual smallpox mortality in London for the period 1629 to 1893 (based on annual Bills of Mortality until 1836 and on the Registrar General's Annual Returns for 1837 to 1893). In Fig 2, we compare these annual counts with annual aggregations of our weekly data.

Differences between the two data sets are indicated with white stacked bars if the annual counts from Creighton are larger, and black stacked bars if our annual sums from the weekly data in this paper are larger. Before 1664, there are no weekly data, so the bars are entirely white. After 1893, we show only our annual sums and color them red.

Annual smallpox mortality was greatest in 1871; our annual sum for that year is 7,982, and Creighton's table indicates 7,912. The top of this bar is cut off in Fig 2. In all other years, there were fewer than 4,000 smallpox deaths.

The data from the two sources align well for most years. The most substantial discrepancies are during the period of transition from the Bills of Mortality to the Registrar General's Returns (1837 to 1841); we summed weekly LBoM data for this period, whereas Creighton

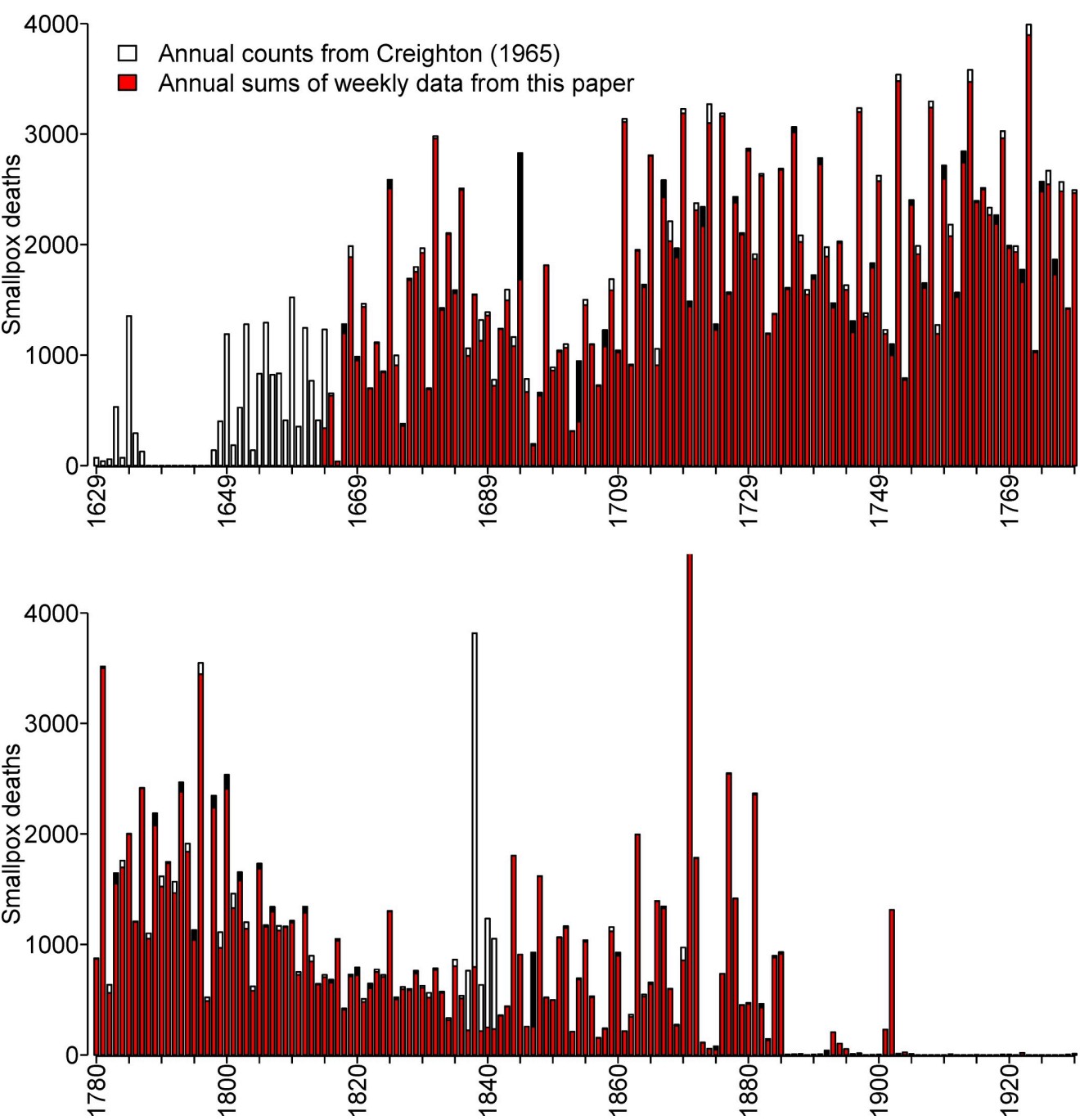

**Fig 2. Consistency of annual smallpox mortality in London, England, as tabulated by Creighton [36] from annual records, and in this paper from weekly records.** Each panel shows 150 years: 1629–1779 (**top panel**) and 1780–1930 (**bottom panel**). Where the tops of stacked bars are white, the annual counts are larger; where the tops are black, the weekly sums are larger. The gap from 1637 to 1646 is due to missing annual bills ([36], p. 437). The data and R script required to reproduce this figure are available at https://github.com/davidearn/London_smallpox.

used the Annual Registrar General's Returns. Creighton notes that the annual count for 1837 includes only six months, not the full year ([36], p. 613), but even so, it exceeds our sum from the weekly bills (which is not surprising given the much larger geographical area covered by the Registrar General's Returns; see Transition to the Registrar General's Weekly Returns section).

**Table 1. Population of London, England (1550–1931).**

| Year | Inner London | Outer London | Total London | Source |
|------|-------------|--------------|--------------|--------|
| 1550 | 120,000 | | 120,000 | [55] |
| 1560 | 140,800 | | 140,800 | [55]* |
| 1580 | 180,400 | | 180,400 | [55]* |
| 1600 | 200,000 | | 200,000 | [55] |
| 1620 | 297,000 | | 297,000 | [55]* |
| 1640 | 390,500 | | 390,500 | [55]* |
| 1650 | 391,450 | | 391,450 | [55]* |
| 1660 | 392,400 | | 392,400 | [55]* |
| 1680 | 468,600 | | 468,600 | [55]* |
| 1700 | 490,000 | | 490,000 | [55] |
| 1735 | 660,000 | | 660,000 | [40] |
| 1745 | 670,000 | | 670,000 | [40] |
| 1750 | 675,000 | | 675,000 | [55] |
| 1755 | 680,000 | | 680,000 | [40] |
| 1765 | 730,000 | | 730,000 | [40] |
| 1775 | 780,000 | | 780,000 | [40] |
| 1785 | 826,502 | | 859,234 | [40] |
| 1795 | 909,507 | | 1,007,703 | [40] |
| 1801 | 959,310 | 137,474 | 1,096,784 | [54] |
| 1811 | 1,139,355 | 164,209 | 1,303,564 | [54] |
| 1821 | 1,379,543 | 193,667 | 1,573,210 | [54] |
| 1831 | 1,655,582 | 222,647 | 1,878,229 | [54] |
| 1841 | 1,949,277 | 258,376 | 2,207,653 | [54] |
| 1851 | 2,363,341 | 288,598 | 2,651,939 | [54] |
| 1861 | 2,808,494 | 379,991 | 3,188,485 | [54] |
| 1871 | 3,261,396 | 579,199 | 3,840,595 | [54] |
| 1881 | 3,830,297 | 883,144 | 4,713,441 | [54] |
| 1891 | 4,227,954 | 1,344,014 | 5,571,968 | [54] |
| 1901 | 4,536,267 | 1,970,622 | 6,506,889 | [54] |
| 1911 | 4,521,685 | 2,638,756 | 7,160,441 | [54] |
| 1921 | 4,484,523 | 2,902,232 | 7,386,755 | [54] |
| 1931 | 4,397,003 | 3,713,355 | 8,110,358 | [54] |

Entries with an asterisk were estimated based on linear interpolation from ([55], Tables 2 and 5).

## London's population

Decennial censuses of London began in 1801 [54]. We estimated London's population at earlier times based on Finlay and Shearer [55] for the period up to 1700, and Landers [40], who provides decadal population estimates from 1735 to 1795 (see Table 1 and the top of Fig 3).

Finlay and Shearer [55] estimated the total population of London for every half century from 1500 until 1700 based on information about the population north and south of the river Thames, taking into account inaccuracies in data collection. Their book [55] also contains data for other years for the population north ([55], Table 2) and south [55] of the river that were not included in their summary ([55], Table 5). Population numbers for some years, for the south of the river, were missing (years highlighted with an asterisk in Table 1); we used linear interpolation to estimate the population south of the river for these missing years. Final total

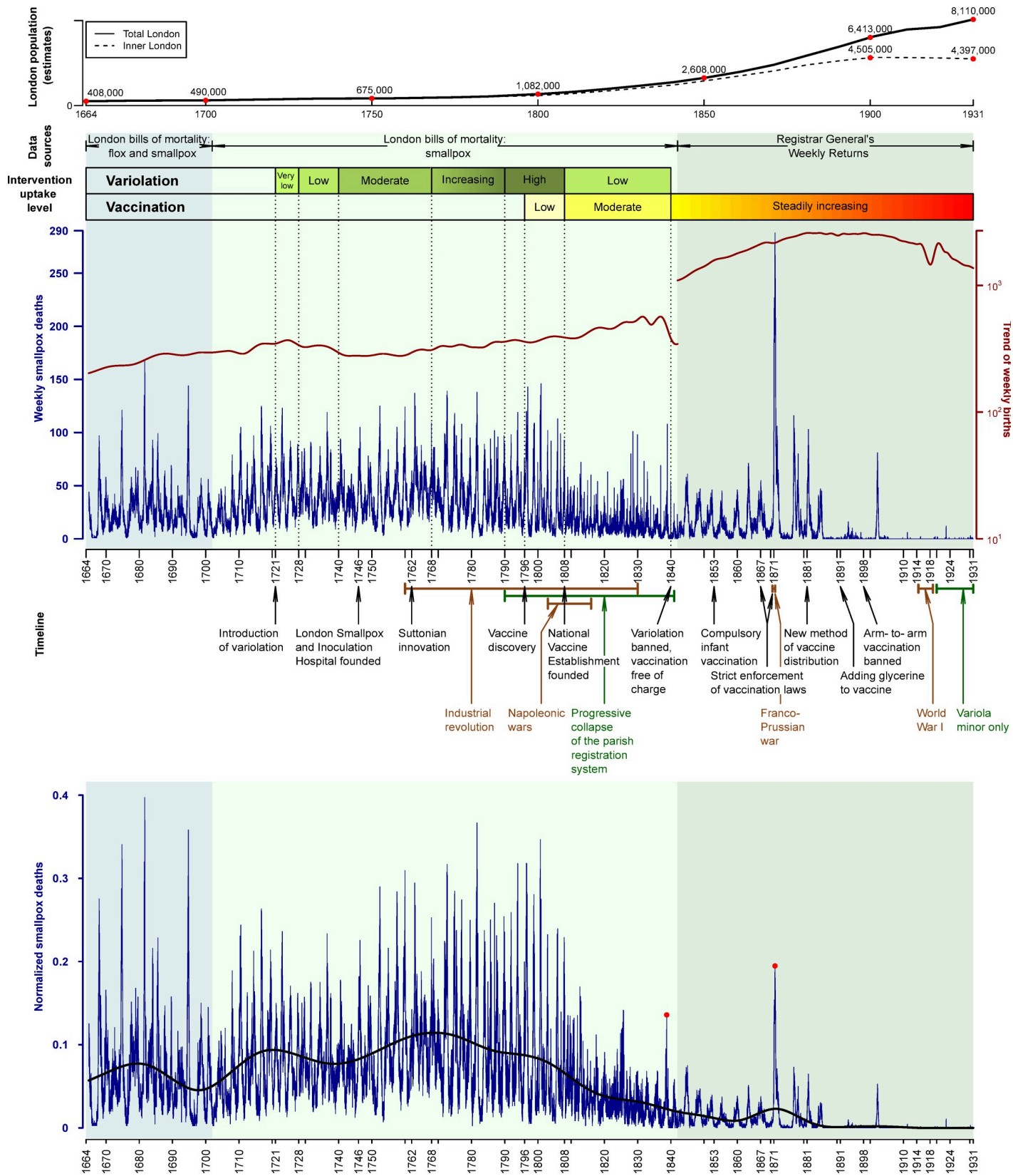

**Fig 3. London's population and weekly smallpox deaths, 1664–1930, against the timeline of historical events related to the history of smallpox in England.** The top of the graph shows the population of inner London (dashed) and all of London (solid) as estimated in the London's population subsection of Data in the main text. The **main panel** shows weekly smallpox deaths in London in blue and the long-term trend of weekly births in red. Background shading indicates different sources and classifications of disease: 1664–1701, flox and smallpox burials from the London Bills of Mortality (LBoM); 1701–1841, smallpox burials from the LBoM; 1842–1931, smallpox deaths from the Registrar General's Weekly Returns (RGWRs). The geographical area covered by the RGWR was larger than for the LBoM (see Data for details). Qualitative variolation and vaccination uptake levels are shown with colored bars. Annotation below the main panel shows the timeline of historical events related to smallpox history in England: Black text indicates events that influenced uptake of control measures; brown text indicates events that influenced human behavior; and dark green text is used to highlight reduced data accuracy during the last few decades of the LBoM. The **bottom panel** shows weekly smallpox deaths normalized by the long-term trend of all-cause deaths (see Fig 4). The trend of normalized smallpox deaths is also shown (heavy black curve). Trends were estimated by Empirical Mode Decomposition (see Methods). Red dots highlight the peaks of the epidemics of 1838 and 1871, the most significant smallpox epidemics of the 19th century. The data and R script required to reproduce this figure are available at https://github.com/davidearn/London_smallpox.

population estimates were calculated by the same method as in [55], i.e., adding the populations north and south of the river and multiplying the sum by an inflation factor (1.1, except 1.2 in 1650 and 1660) to account for inaccuracy in data reporting. Note that Finlay and Shearer's estimate for 1650 appears to be inaccurate due to an arithmetic error (according to their own figures, it should be approximately 400,000, not 375,000).

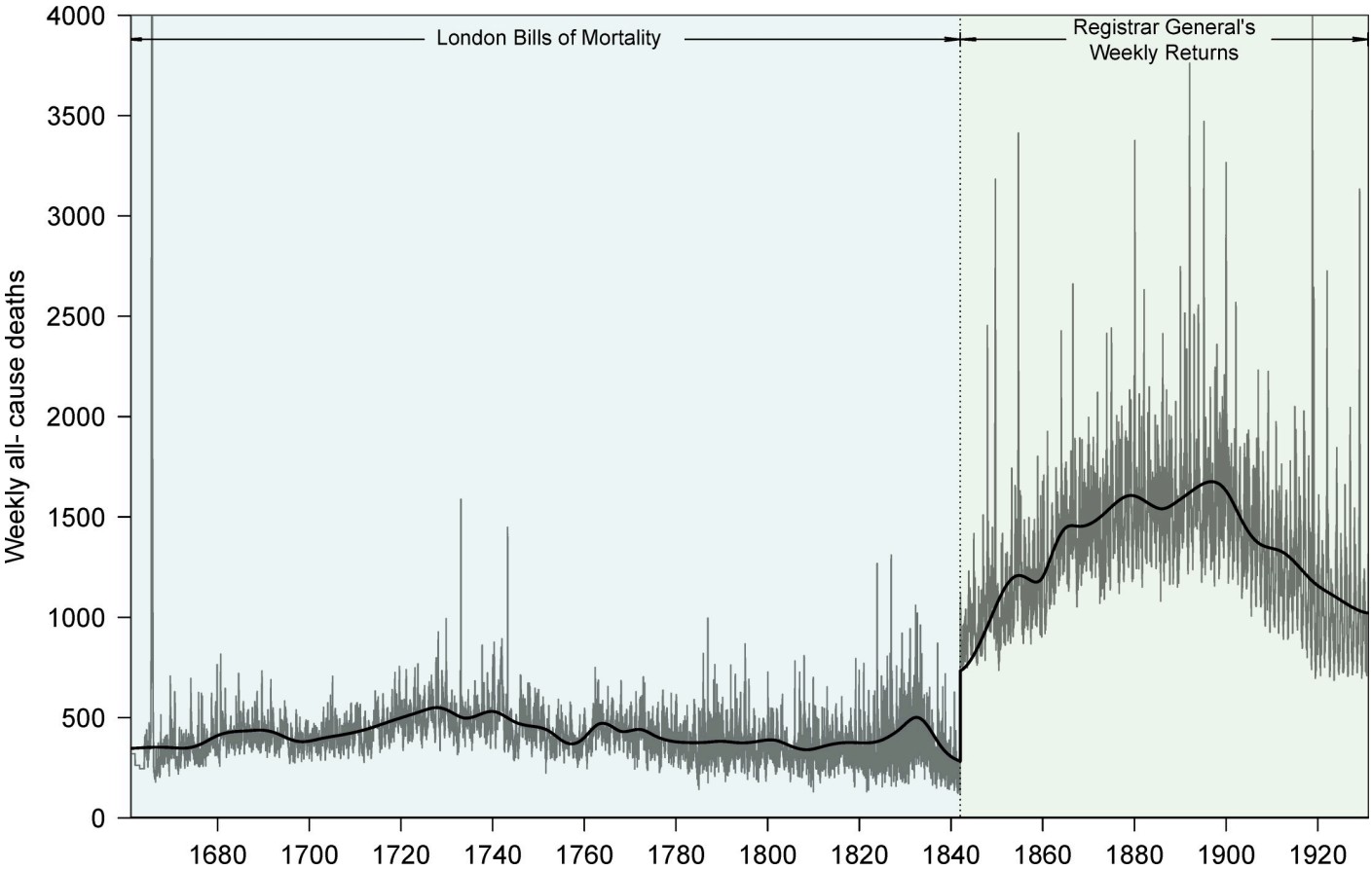

**Fig 4. Weekly all-cause deaths in London, England, 1661–1930.** The trend was estimated by Empirical Mode Decomposition (see Methods) applied separately to the periods 1661–1841 and 1842–1930, which correspond to different data sources. Two peaks are cut off in the plot. During the Great Plague of London, deaths reached 8,297 per week in September 1665. During the 1918 influenza pandemic, deaths reached 4,290 per week in November 1918. The data and R script required to reproduce this figure are available at https://github.com/davidearn/London_smallpox.

## Annotation of data with historical events

Over the 267 years that we study, London underwent major demographic and social changes, and there were a variety of historical events that may have had substantial impacts on smallpox dynamics. The introduction of smallpox control measures (variolation and later vaccination) would be expected to influence smallpox dynamics. Other events that could potentially have impacted smallpox epidemics include wars [56,57] and the Industrial Revolution, which was accompanied by urbanization and demographic transitions [58–61]. We annotate the small-pox time series in Fig 3 with the major events and developments that we describe below.

## Control-free era

The first recorded outbreak of smallpox in England is dated to 12 August 1610 ([36], p. 435). The first annual Bill of Mortality (1629) recorded only 72 burials due to smallpox, but later, bills often listed many more; in 1634, for example, the annual Bill recorded 1,354 smallpox burials ([36], p. 436). The cumulative death toll from smallpox over the course of the 17th century eventually exceeded plague, leprosy, and syphilis [2,35,40]. According to the annual Bills of Mortality, there were approximately 97,000 burials due to smallpox between 1629 and 1721 (excluding 1637 to 1646, for which there are no extant mortality bills) [36].

## Early variolation, 1721 to 1740

Lady Mary Wortley Montagu (an influential writer and poet [62]) is credited with introducing variolation to Great Britain [1–3,30,63]. She had her daughter professionally variolated in London in April 1721 (this year is annotated with "Introduction of variolation" in Fig 3). For the next two decades, variolation occurred but was not popular. Only 857 people were variolated in the whole of Great Britain from 1721 to 1727 and only 37 in 1728 [36]. The variolation level for this period is consequently indicated as "Very low" in Fig 3. In the decade following 1728, the frequency of variolation is unknown, but we assume it was between the very low level before 1728 and the moderate level later when it became a more common practice (in Fig 3, the period 1728 to 1740 is indicated as "Low" variolation levels). English medical practitioners implemented variolation very crudely with deep incisions that caused severe symptoms, morbidity, and high mortality of up to 2% ([64], p. 464) [65].

## Increasingly common variolation, 1740 to 1808

Variolation became more popular in the 1740s (indicated by "Moderate" uptake levels in Fig 3) ([64], p. 466; [66], p. 4; [67], p. 17; [68], p. 18). Initially, only the rich could afford it, but charitable variolation began with the establishment of the London Smallpox and Inoculation Hospital in 1746 (annotated in Fig 3).

Until 1762, variolation was usually preceded by four to six weeks of preparation, which included purging, bleeding, and a restricted diet with limited quantities of food. Variolation was followed by an isolation period of two or more weeks. Isolated individuals were placed in purpose-built inoculation houses ([1], [3], p. 255). The preparation period was shortened after Robert Sutton's improvement of variolation using light incisions in 1762 (annotated in Fig 3). Sutton's new method dramatically decreased the severity of symptoms and death and reduced the cost. Consequently, it became more common to offer variolation to the poor population free of charge.

Sutton's variolation technique spread quickly around England and became very popular in rural areas. When a new epidemic appeared to be highly probable, "general variolation" of entire villages and communities was performed. In large towns and cities, the situation was

quite different [25]. In London, the use of variolation was irregular and attempts to perform "general variolation" were sporadic and rare.

Poor Londoners were variolated only through Smallpox Charities. They performed public variolation in batches, separately for males and females, 8 to 12 times a year ([36], p. 506). The charities did not admit children under 7 years of age despite the fact that the vast majority of smallpox cases at that time were in infants and young children ([36], p. 507).

The full extent of variolation in London after the Suttonian innovation is unclear. The figures from London Hospital show that the number of variolated individuals increased dramatically from 29 in 1750, to 653 in 1767, and 1084 in 1768 ([36], p. 506). Based on a variety of historical reports, Razzel concluded that variolation gained considerable popularity in London at the turn of the 19th century ([3], p. 72). From these qualitative descriptions, it seems likely that uptake of variolation increased after 1768 and reached a maximum during 1790 to 1808 [3,69] (annotated in Fig 3).

## Industrial revolution

During the Industrial Revolution (annotated as 1760 to 1830 [70] in Fig 3), a "growing number of people moved to urban centres" [71] such as London, precipitating significant demographic and social changes [38,58–61]. London's population more than doubled from about 730,000 in 1765 to about 1,900,000 in 1831 (see Table 1 and section on London's population). If this increase in population size was associated with an increase in population density, and/or an increase in the average number of contacts people had, then the rate of transmission of small-pox would have increased and affected epidemic patterns [72]. Transmission dynamics would also have been affected by changes in fertility [13,15], which may [38,60] or may not [58] have increased during the Industrial Revolution. It has also been proposed that smallpox transmission increased as a result of viral evolution during this period [23–25].

## Vaccination

The idea of vaccination was initially met with skepticism by the scientific and medical communities [1,2]. Jenner "was advised not to send a record of his observations to the Royal Society, which was prepared to refuse it, but to publish it as a pamphlet; and as a pamphlet it appeared in 1798" ([73], p. 62).

Unlike variolation, vaccination came with relatively little risk to the vaccinee, no preparatory period, and much lower cost. Consequently, in spite of the initial skepticism, vaccination was adopted by the public more quickly and more widely than variolation ever was [1]. There were initially many impediments associated with ineffective vaccine distribution and storage, shortage of cowpox virus, inadequate vaccine efficacy, waning immunity (hence a need for periodic revaccination), and religious and philosophical objections. These challenges were overcome over the course of the 19th century, and by the turn of the 20th century, vaccine uptake had risen sufficiently to cause a dramatic decline in smallpox mortality (Fig 3).

Unfortunately, quantitative reports of early smallpox vaccine uptake are sorely lacking. Vaccinations were poorly recorded until the end of the 19th century. Available data are incomplete, uncertain, and inconsistent. For example, the figures from the London Smallpox and Inoculation Hospital show the percentage of vaccinated patients admitted to the hospital increasing steadily from 32% in 1825 to 73% in 1856 [74], whereas the Royal Commission on Vaccination found that only 25% of newborns were vaccinated by 1820 and about 70% in some parishes by 1840 [35]. Mooney [75] states that during 1854 to 1856, the percentage of vaccinated infants might have ranged from 28% to 81% based on one source, but that another source indicates that infant vaccination rates for London during the period 1845 to 1890 were much lower than the national average and never increased above 500 per 1,000 live births (i.e.,

50%). There appear to be no surviving records concerning vaccinations of older age groups during this period; however, it is hypothesized that many adults escaped vaccination ([74], p. 117). In Fig 3, we indicate the qualitative pattern of change in vaccine uptake.

### Better access and increasingly strong legislation concerning vaccination

A number of additional developments and policy changes during the 19th century are indicated in the timeline in Fig 3.

State involvement in the control of smallpox in England began with the foundation of the National Vaccine Establishment in 1808. It provided free vaccination at its London stations and distributed vaccine to other parts of England [76]. Around this time, the London Smallpox and Inoculation Hospital ceased variolation and began vaccination in greater numbers. Smallpox outbreaks over the next decade were very mild, possibly resulting from increased vaccination; however, this was precisely the period of the Napoleonic Wars (1803 to 1815) during which other infectious diseases were also less prevalent than usual in England ([36], p. 569).

A strikingly large smallpox epidemic occurred in London from 1837 to 1838 and exploded into a European wide pandemic [2]. The authorities in England realized that some radical measures had to be taken, which led to the first Vaccination Act of 1840, providing vaccination free of charge and banning variolation. It was followed by the Vaccination Act of 1853, which made vaccination of every child during the first four months of life compulsory. The Vaccination Act of 1867 introduced penalties for not complying with compulsory vaccination.

The Franco–Prussian war, which began in late July 1870, is believed to have initiated the worst pandemic of smallpox in all of Europe in the 19th century. It resulted in at least 500,000 deaths. England alone lost more than 40,000 people. Thanks to compulsory vaccination, fatality rates in England were three times lower than in Prussia, Austria, and Belgium ([2], pp. 87–91). The immediate response of the English government to this devastating pandemic was the Vaccination Act of 1871, which enforced very strict control (through the courts) of the implementation of the previous Acts.

In the second half of the 19th century, many vaccine-related challenges were resolved. Arm-to-arm vaccination (vaccine transfer from the infectious pustule of vaccinated individual to a nonvaccinated individual [1]) was the main method of vaccine distribution in the beginning of the century. It was dangerous because it could transmit other diseases such as syphilis and was consequently outlawed in 1898 [2]. It was replaced by the new technique of passing cowpox from cow to cow. The new method of vaccine distribution was first introduced in Naples, Italy, in 1843 [77,78]. However, it arrived in England only in 1881. Another important discovery was made in 1891 by Monckton Copeman [78], who demonstrated that adding glycerine to smallpox vaccine reduces bacterial contamination, making it more efficacious and reliable.

### Eradication in the 20th century

The last large outbreak of *Variola major* occurred in London from 1901 to 1902 (Fig 3) and was probably seeded from another country [74]. After 1902, only very small outbreaks occurred with very low incidence and very few deaths.

In 1967, the World Health Organization launched its global smallpox eradication campaign. In 1980, smallpox was certified as the first infectious disease to be eradicated by human efforts [27].

### Methods

We used a variety of methods to elucidate the patterns in the London smallpox time series. Spectral analyses and a visualization of seasonality allow us to reveal important characteristics of the data that cannot be gleaned by inspection of the raw time plot (Fig 3).

## Normalization

Due to population growth and changes in city boundaries, the size of the population from which smallpox deaths were reported for London changed over the course of the time series. In addition, sampling of deaths in the LBoM was less complete in the last few decades before the establishment of the RGWR. Sampling deficiencies presumably affected deaths from all causes in the same way; consequently, we attempted to obtain a consistent normalization of weekly smallpox deaths by dividing by the trend of weekly all-cause mortality (ACM) in London, rather than estimated population size. To calculate the trend in ACM, we used **Empirical Mode Decomposition (EMD)**, which is designed to identify trends in nonlinear and highly nonstationary time series [79,80,81]. EMD was developed to overcome the drawbacks of moving averages, other linear filters, or linear regression, which often perform poorly on nonstationary data. EMD decomposes a signal into several components with a well-defined instantaneous frequency via **intrinsic mode functions (IMFs)**. IMFs are basically zero-mean oscillation modes present in the data: The first IMF captures the high frequency (shorter period) oscillations, while all subsequent IMFs have lower average frequency (longer period). Each IMF is extracted recursively starting from the original time series until there are no more oscillations in the residue. The last residual component of this process can be considered an estimate of the trend [80].

## Spectral analysis

We used spectral analyses to identify the strongest periodicities in the smallpox time series, both globally (with a traditional Fourier analysis) and locally (via wavelet analysis). Before computing spectra, we normalized, square-root transformed, and detrended the data in order to reduce variation in amplitude without affecting periodicities [82,83].

**Classical power spectrum.**   We computed the standard **power spectral density** [14,84–86] of the entire smallpox time series to obtain a global estimate of its frequency content. We used the R [87] function `spec.pgram` with no taper (taper = 0) and a standard modified Daniell smoother (`kernel = kernel("modified.daniell",c(3,3))`).

**Wavelet spectrum.**   Because infectious disease time series are typically highly nonstationary, wavelet analysis has become increasingly popular in epidemiological research [16,82,83,88–90]. We computed a wavelet transform [91,92] of the smallpox time series in order to examine how smallpox periodicities changed over the course of the three centuries.

A wavelet transform is computed with respect to a basic shape function, the **analyzing wavelet**, which has a **scale parameter** that controls its width. Narrower (wider) scales correspond to higher (lower) frequency modes in the time series. At any given time and scale, a stronger correlation between the analyzing wavelet and the signal yields a larger value of the wavelet transform. We obtain a complete (two-dimensional) time-frequency representation of the data by convolving the analyzing wavelet—at each scale—with the original time series. We used the Morlet wavelet [92] as the analyzing wavelet to obtain the wavelet transform of the smallpox time series.

The standard computation of the wavelet spectrum requires that the number of points in the time series be a power of 2. Consequently, we pad the ends of the time series with zeros to bring the number of time points in the data to the nearest power of 2. The resulting artificial discontinuity where the zero-padding begins reduces the accuracy of the wavelet transform at the ends of the time series. Regions of lower accuracy are identified by the **cone of influence**. Data outside this cone should be interpreted with caution. Ninety-five percent confidence regions are computed based on 1,000 Markov bootstrapped time series [83,89].

## Seasonality

An indication of underlying seasonality in a time series is the occurrence of a spectral peak at a period of one year. However, this crude measure suppresses the detailed seasonal pattern and, in particular, does not reveal the times of year when peaks or troughs occur.

Following Tien and colleagues [7], we visualized the evolving seasonal pattern of smallpox dynamics with a heat map in the time-of-year versus year plane (we refer to this as a **seasonal heat map**).

Before constructing the seasonal heat map, we square-root transform and detrend the normalized smallpox time series (because we have found that seasonality is represented mostly clearly after this transformation). Detrending has the effect of shifting the local mean at a given time toward the global mean of the entire time series. Consequently, the last few decades of the smallpox time series, which have a true mean close to zero, are represented by "medium heat" in the seasonal heat map because the true zeros are shifted to the global mean.

To further clarify changes in smallpox seasonality, we separately identified the peak week of each epidemic by visual inspection of the normalized time series, and displayed all peak weeks in the time-of-year versus year plane. To assist with identification of trends, we also computed a moving average of the week-of-the-year in which an epidemic peak occurred (using a circular average that considered week 53 and week 1 to be the same). Our moving average was calculated with a 21-epidemic window, i.e., each plotted point is the average of the previous 10, current, and next 10 epidemic peak weeks (peaks did not necessarily occur every year).

## Results

### Patterns in the raw time series

The time plot of the raw data (Fig 3, top panel) displays substantial changes in the structure, amplitude, and frequency of smallpox epidemics over time. However, some of the apparent changes are misleading because they do not account for population growth and inconsistency of data sources. For example, the epidemic of 1871 appears to be the largest, but it was not the largest relative to population size or relative to ACM. In contrast, the epidemic of 1838, which is frequently mentioned in the literature [2,36], is not easily identifiable in the raw data, likely because it occurred just at the time of transition between the LBoM and the RGWR.

### Normalization

Weekly ACM in London (1661 to 1930) and the EMD-computed trends for the periods 1661 to 1841 and 1842 to 1930 are presented in Fig 4. The large difference in population coverage (see Data section) between the LBoM and RGWR is evident.

### Patterns in the normalized time series

The bottom panel of Fig 3 shows the weekly London smallpox time series normalized by the ACM trend.

The raw and normalized time series emphasize different features of smallpox dynamics in London. For example, in the bottom panel of Fig 3, the epidemic of 1838 is easily identifiable, and the epidemic of 1871 is much less extreme in magnitude compared with other epidemics of the 19th century.

From the earliest times in the series, there were recurrent epidemics. Four large outbreaks in 1667, 1674, 1681, and 1694 stand out in the normalized time series, and an epidemic of this magnitude did not occur again until 1752. The epidemic pattern before about 1705 appears to

have been less stationary than the pattern in the subsequent decades (the frequency and amplitude of outbreaks was more consistent from 1705 to 1750).

The years from 1770 to 1810 were characterized by stricter regularity of epidemics. This period coincided with more common variolation (the practice gained popularity after the Suttonian innovation of 1768). Beginning around 1810, the data show a dramatic reduction in the amplitude of epidemics, though outbreaks were more frequent and the data are noisier. The declining trend in epidemic severity is temporally associated with the introduction of vaccination; unfortunately, this was precisely the period over which the parish registration system collapsed, increasing the difficulty of estimating the true impact of vaccination in the early vaccine era.

After 1835, interepidemic intervals increased and (normalized) epidemic peak heights declined, with the exception of four large epidemics in 1837, 1871, 1876, and 1902. During this period, variolation was eliminated and vaccination levels increased. The coincidence of the 1840 Vaccination Act (which made variolation illegal) with the radical change in data collection from the LBoM to the RGWR limits our ability to identify potential causal links between policy and behavioral changes and epidemic patterns.

The bottom panel of Fig 3 also shows the EMD-computed trend of normalized smallpox mortality in London over the centuries. As a proportion of ACM, smallpox mortality rose steadily from 1664 until 1680 and then fell until about 1700. After this drop, the trend generally increased (with a shallow dip around 1740) until approximately 1770. After 1770, the trend declined gradually until 1908, when smallpox deaths were nearly eliminated in London.

After 1840, epidemics became more regular with a longer interepidemic period of 3 to 4 years. After the exceptionally large epidemic of 1871 (there were 10,618 reported smallpox deaths from 1 January 1870 to 31 December 1872), subsequent smallpox outbreaks were much smaller and irregular. The last substantial outbreak was in 1902, after which there were very few smallpox deaths reported.

## Spectral analysis

**Classical power spectrum.** Fig 5 shows the period periodogram (power spectrum as a function of period) for the full smallpox time series. A strong peak at one year suggests underlying seasonality of epidemics. Other peaks (near 2.2, 2.4, 3, 5.1, and 6 years) suggest more complex dynamical patterns.

**Wavelet spectrum.** Fig 6B shows the wavelet transform of the normalized London smallpox time series. Colors indicate the strength of signal at given periods (blue meaning weak and red meaning strong). The cone of influence and 95% confidence contours are shown in black.

The black curves through the bright areas of Fig 6B indicate the periods with greatest power at each time point. From 1664 to about 1720, multiple spectral modes are prominent (initially periods near 2, 3, and 5 years, then near 3 and 5 years only from 1690 to about 1705, and then near 2 years only until about 1728). There is a weak signal between 2 and 3 years from 1728 to 1740, and then a strong signal near 2 years (1740 to about 1765) and near 3 years (about 1770 to 1808). A relatively weak spectral peak at one year can be seen over much of the time series before 1820, though its magnitude is below the threshold for drawing a black peak line except for the decade from 1798 to 1808.

From about 1808 to about 1832, the time series is markedly noisier, displays much lower amplitude fluctuations, and much weaker spectral signals. The decay of the parish registration system may have contributed to this temporal pattern, though the clear recurrent epidemic pattern that emerges about 1832—well before the beginning of the RGWR era—suggests that there were genuine changes in smallpox dynamics at the end of the LBoM era, not just a change in sampling.

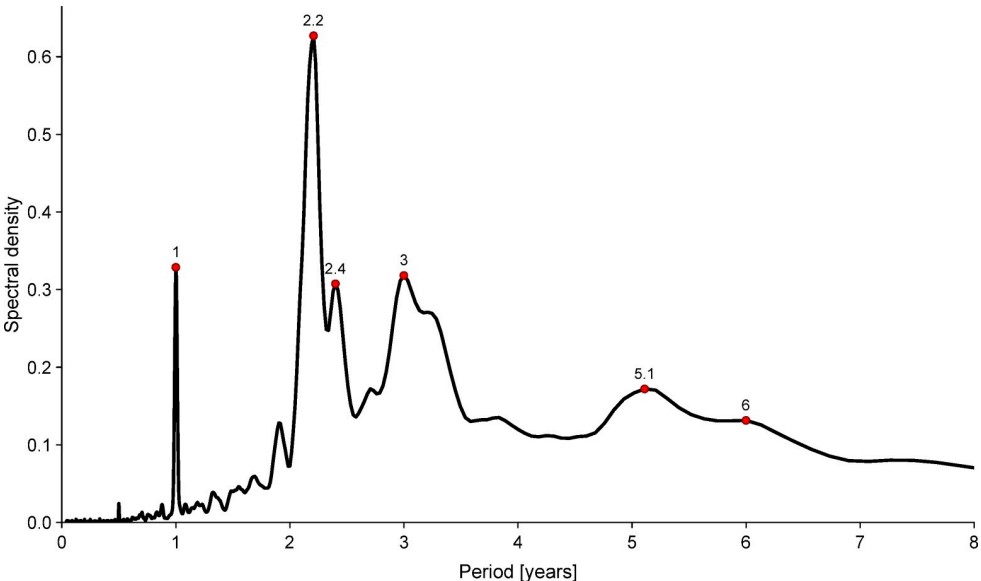

**Fig 5. Classical Fourier power spectrum of the normalized weekly smallpox mortality time series for London, England, 1664–1930.** Before computing the power spectrum, the time series was square-root transformed and detrended [84]. A standard modified Daniell smoother was used in the computation of the spectrum (see Methods). The data and R script required to reproduce this figure are available at https://github.com/davidearn/London_smallpox.

## Seasonality

Fig 6C shows the seasonal heat map for smallpox mortality in London, and Fig 6D shows all peak weeks in gray and the moving average peak week (computed from the 21 nearest outbreaks) in red. The seasons are indicated approximately with horizontal dashed lines: Winter (week 1, the first week of January), Spring (week 13, end of March), Summer (week 26, end of June), and Fall (week 39, end of September).

Reported smallpox mortality was lowest in the spring until about 1840, and in the fall/winter afterwards (these are the bluest regions in Fig 6C). The time series is generally very noisy during the troughs between epidemics, and there is rarely a clear minimum week.

Around 1750, there was a shift in the typical timing of outbreak peaks from summer/fall to fall/winter. This change is clearest in the red moving average in Fig 6D. Epidemic patterns might have become less seasonal during the period from 1808 to 1840 (the range of "temperature" is narrower in the heat map in this segment of Fig 6C). After 1840, as epidemics became less frequent, they peaked in the winter/spring.

## Discussion

We have digitized and analyzed what is, to our knowledge, the longest existing weekly time series of infectious disease mortality. The time span of the data, from 1664 to 1930, covers an extraordinary period in London, England, during which smallpox changed from a terrifying and unavoidable danger to an easily preventable infection.

### Previous work on smallpox dynamics in London

A number of previous studies of smallpox in London are complementary to the analysis presented here.

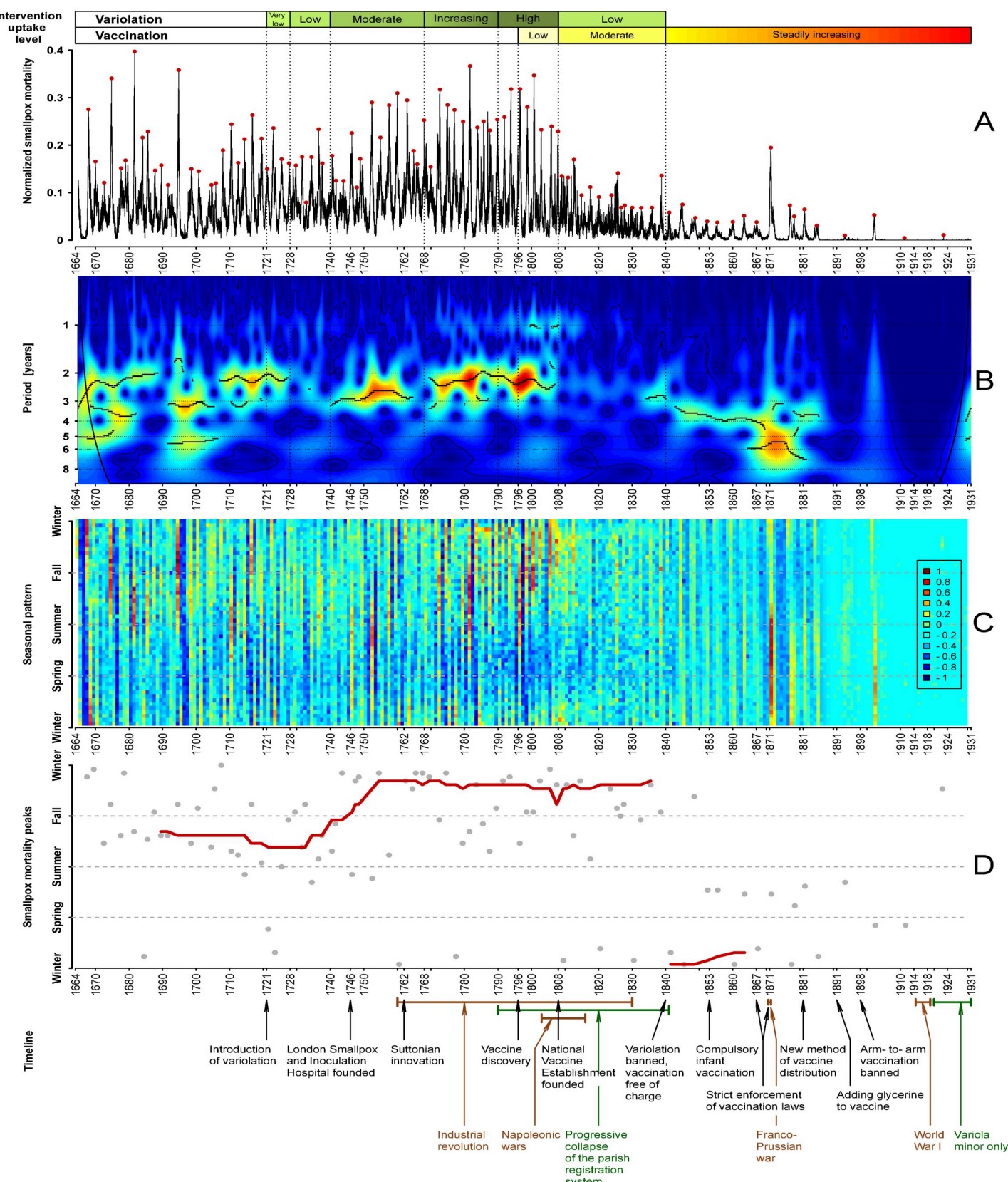

**Fig 6. Spectral structure and seasonality of smallpox dynamics in London, England, 1664–1930. Panel A:** weekly smallpox deaths normalized by the long-term trend of all-cause deaths. The red dots indicate the epidemic peaks, i.e., the highest value of normalized smallpox mortality during the epidemic year (identified by visual inspection). **Panel B:** wavelet transform of the normalized weekly smallpox mortality time series (square root-transformed and normalized to unit variance). Colors range from dark blue for low power to dark red for high power. Heavy black curves show the local maxima of wavelet power (squared modulus of wavelet coefficients [89]) at each time. Thin black curves show 95% confidence contours, estimated from 1,000 bootstrapped time series generated by the method of [89]. Medium black curves near the left and right edges show the **cone of influence** [89,92], below which the calculation of wavelet power is less accurate because it includes edges of the time series that have been zero-padded to make the number of time points a power of 2. The wavelet spectrum was computed using MATLAB code kindly provided by Bernard Cazelles. **Panel C:** seasonal heat map based on the detrended, square-root transformed, normalized smallpox time series (see Seasonality subsection of Methods). Colors indicate the degree of variation from the global mean of the time series (which is 0 after detrending). **Panel D:** The epidemic peak times (red dots in Panel A) are displayed with gray dots in the year versus time-of-year plane. The red curve shows the moving average peak week, computed using a window of 21 epidemics. As in Fig 3, variolation and vaccination uptake levels are indicated above Panel A, and the timeline of historical events related to smallpox history in England is shown below Panel D. The data and R script required to reproduce this figure are available at https://github.com/davidearn/London_smallpox.

In the 1990s, Duncan and colleagues [21,93,94] studied annual smallpox mortality in London, from 1647 to 1893, and estimated interepidemic intervals in five segments of the time series ([21], Table 1) (these results are reproduced in Fig 7). These authors attributed changes in interepidemic intervals to population growth, malnutrition associated with changes in wheat prices, and seasonal variations in temperature and humidity. Cliff and colleagues ([48], p. 101) have also estimated interepidemic intervals for smallpox in London using traditional time series analysis (also reproduced in Fig 7). For comparison, in the lower panel of Fig 7, we show the interepidemic intervals inferred directly from the time between adjacent peaks in the smallpox time series.

More recently, Davenport and colleagues [23–25] have examined individual, age-specific death records in a large area of London over a period of several decades (1752 to 1805). These authors discovered that, after 1770, smallpox mortality declined in adults and rose in children. They suggested that this change in the age distribution of smallpox mortality might have been a consequence of evolution of increased transmissibility of the *Variola* virus.

## New data and descriptive statistics

Digitization of the full weekly record of mortality from smallpox (and from all causes) in London has enabled us to conduct a number of informative analyses that reveal changes in the seasonal structure of smallpox epidemics over the centuries. The wavelet spectrum (Fig 6B) provides a more detailed view of the periodic structure of smallpox than has been accessible previously. In addition, our seasonal heat map (Fig 6C) and epidemic peak time visualization (Fig 6D) show how the seasonal timing of outbreaks varied over many decades.

The annual data that have been studied previously smooth out the seasonal patterns that we have identified and restrict the resolution of spectral features. Even the classical power spectrum of the weekly data (Fig 5) reveals noninteger periods that cannot be detected from annual counts, and the wavelet spectrum (Fig 6B) shows gradual changes in periodic structure. In Fig 7, we compare the spectral peaks from our wavelet spectrum with the interepidemic intervals previously estimated from annual data.

## Interpretations

The primary role of vaccination in the ultimate elimination of smallpox mortality in London is not in question. However, the causes of the various transitions in the spectral and seasonal structure of smallpox dynamics over the decades and centuries are not clear.

The historical timelines of events and uptake of interventions, shown alongside the smallpox data and analyses in Fig 3, should aid in formulating mechanistic hypotheses, especially in relation to the history of control strategies. Previous work has shown that long-term changes in birth rates and vaccination levels have induced transitions in transmission dynamics of

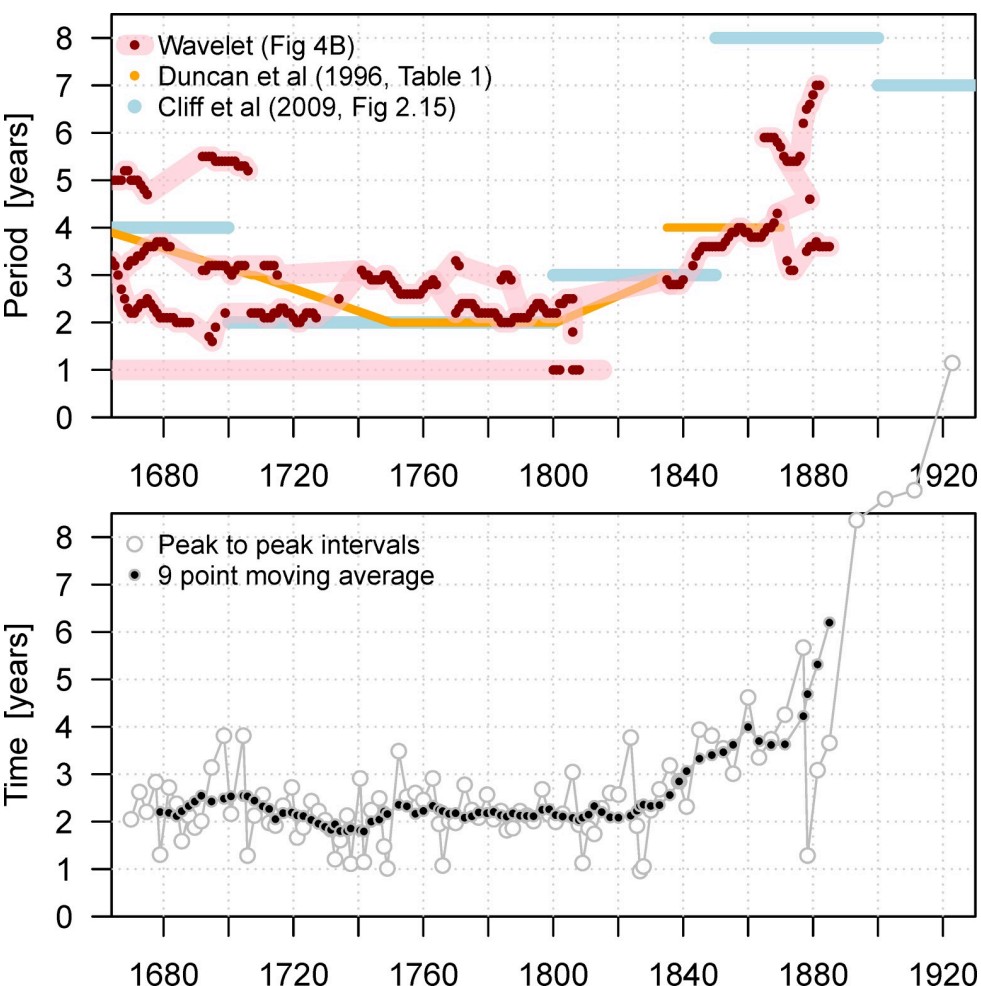

**Fig 7. Periodicities in the time series of smallpox mortality in London, 1664–1930.** Upper panel: primary spectral peaks, as estimated in previous work [21,48] (based on traditional spectral analysis of annual data), and in this paper (based on a wavelet analysis of weekly data). For the wavelet analysis, all spectral peaks above the threshold used in Fig 6B are shown with red dots; the pink curves underneath are based on visual identification of peaks in Fig 6B. Lower panel: Interepidemic period estimated by the time between successive epidemic peaks. A nine-point (central) moving average of the peak-to-peak intervals is also shown. The data and R script required to reproduce this figure are available at https://github.com/davidearn/London_smallpox.

other infectious diseases [13–16]. Time series of additional historical and environmental variables are also important to consider. Economic factors, such as wheat prices [21], might have affected transmission and/or mortality rates. Weather changes, especially in humidity and temperature, may also have influenced seasonality of smallpox [21,40], and climatic changes (see Fig 1 of [95]) could have influenced periodic dynamical structure over longer timescales.

More information about the age, social, and spatial structures of smallpox mortality would be extremely valuable. A case in point is the discovery by Davenport and colleagues [23–25] that, around 1770, smallpox burials declined in adults and rose in the very young. This shift in age structure of mortality seems likely to have been driven by a corresponding shift in the age structure of infections. The most obvious potential cause of a trend toward younger age at infection is an increase in the transmission rate [72]. Davenport and colleagues suggest that greater transmission may have resulted from "a sudden increase in infectiousness of the smallpox virus" ([23], p. 1289). Alternatively, the virus could have evolved to yield a longer infectious period.

The plausibility of unusual viral evolution around 1770 is supported by recent molecular genetic analyses, which found that "diversification of major viral lineages only [occurred] within the 18th and 19th centuries, concomitant with the development of modern vaccination" [96]. It is certainly possible that evolutionary pressures resulting from changes in control strategies could have selected for increased transmission [97,98]. Of course, a variety of other factors could also have contributed to a rise in the transmission rate. In particular, more common variolation could have promoted transmission (even if it reduced mortality), and population density might have risen during the Industrial Revolution, as mentioned above.

## Seasonality

Any or all of the sociodemographic, behavioral, and environmental factors mentioned above might have contributed in some way to the dramatic changes in the time of year when smallpox mortality peaked over the centuries (Fig 6C and 6D). Recurrence of outbreaks does not necessarily imply that transmission dynamics are seasonally driven. However, the wavelet spectrum in Fig 6B shows a peak at one year from the beginning of the time series until the early 19th century, and the strong spectral peak at exactly one year in Fig 5 suggests underlying exogenous forcing by strictly seasonal variables (e.g., weather, holidays, or migrations during harvests).

If smallpox transmission was seasonally forced, then the implications for dynamical interpretations of the time series are significant. Seasonal forcing of infectious disease transmission can stimulate persistence of complex epidemic cycles [99] and chaos [100]. The mechanistic origin of seasonal forcing may [101] or may not [102] be easy to determine. In the case of smallpox, previous work [1,23,103] found that, in temperate climates, the majority of smallpox cases occurred in the winter and spring, whereas in tropical climates, the seasonality was less pronounced. The general conclusion was that smallpox incidence always increases when the weather is cool and dry; this belief influenced the planning of the eradication campaign in India and seemed to help to improve its efficiency [1]. Previous smallpox seasonality studies were mainly based on data from the 19th and the 20th centuries, when preventative measures were already common [1,103]. Our data set is of a particular interest in this respect since it includes a period when only naturally acquired smallpox immunity existed.

## Explaining transitions in smallpox dynamics

Our goal in this paper has been to describe—and make publicly available—the weekly time series of smallpox mortality in London, and to present information on a variety of historical factors that might have influenced smallpox dynamics over the centuries. The annotated data (Fig 3) will help to frame hypotheses about the mechanisms that were responsible for the observed smallpox mortality patterns, including transitions from one type of pattern to another. Our spectral and seasonal analyses (Figs 5–7) quantify transitions in smallpox dynamics that should be possible to explain using mechanistic mathematical models [72,86,104,105].

Dynamical transitions in measles [13,16] and other childhood infections [14] during the 20th century have been successfully explained with mechanistic models, using observed changes in susceptible recruitment (through changing birth rates and vaccination levels) to predict changes in the frequency structure of epidemic patterns [13–16,86]. For historical smallpox in London, we do have relevant birth data (shown in the top panel of Fig 3), but only qualitative information about vaccination levels (and earlier variolation levels, the effects of which are not entirely clear). Moreover, the time series is so long that the underlying pattern of seasonal forcing probably changed substantially. Changes in seasonal forcing can be accommodated in predictions [16,106], and estimation of underlying seasonal variation in

transmission is becoming easier [107]; however, obtaining a convincing mechanistic explanation for all the structure we have identified in London's smallpox dynamics represents a major challenge.

Ultimately, we anticipate that meeting this challenge will involve further developments of the existing theory of transitions in epidemic patterns [13–16,106], coupled with state-of-the-art methods of inference to obtain parameter estimates for dynamical models [108–112].

## Conclusions

"The greatest killer" [2] has not circulated since its eradication in 1980. While smallpox research in recent years has understandably tended to focus on the potential for accidental or intentional reintroduction in the future, it is enlightening to look back in time. The long history of documenting smallpox mortality in London provides an extraordinary opportunity to learn from the past about changing patterns in infectious disease transmission.

Much remains to be understood about the data we have presented in this paper. From an ecological perspective, the key challenges are to explain the observed smallpox dynamics in London as consequences of intrinsic nonlinear interactions, influenced by identifiable extrinsic forces.

Better control naturally led to less smallpox mortality over time. However, how interventions influence the frequency structure and seasonality of epidemic time series over decades and centuries is much more subtle [13–15]. While preliminary work has been promising [113], careful estimation [114,107] and analysis [15,106] of patterns of seasonal forcing will be required in order to use mechanistic models reliably to explain [16] the observed transitions in smallpox transmission dynamics.

## Acknowledgments

The data were photographed and/or entered primarily by Kelly Hancock, Claire Lees, James McDonald, Laxmi Pandit, and David Richardson. Valerie Hart facilitated work at the Guildhall Library, City of London. Bernard Cazelles shared his code for computing wavelet spectra. We thank all members of the Mathematical Biology Group at McMaster University for helpful comments and discussions.

## Author Contributions

**Conceptualization:** David J. D. Earn.

**Data curation:** David J. D. Earn.

**Formal analysis:** Olga Krylova, David J. D. Earn.

**Funding acquisition:** Olga Krylova, David J. D. Earn.

**Investigation:** Olga Krylova, David J. D. Earn.

**Methodology:** Olga Krylova, David J. D. Earn.

**Project administration:** David J. D. Earn.

**Software:** Olga Krylova.

**Supervision:** David J. D. Earn.

**Visualization:** Olga Krylova, David J. D. Earn.

**Writing – original draft:** Olga Krylova, David J. D. Earn.

Writing – review & editing: Olga Krylova, David J. D. Earn.

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
