## [Editor Report · Decision Letter 0]

9 Sep 2019

Dear Dr Earn, 

Thank you for submitting your manuscript entitled "Patterns of smallpox mortality in London, England, over three centuries" for consideration as a Research Article by PLOS Biology.

Your manuscript has now been evaluated by the PLOS Biology editorial staff, as well as by an academic editor with relevant expertise, and I'm writing to let you know that we would like to send your submission out for external peer review.

IMPORTANT: Although you have submitted your paper as a full Research Article, we think that it would be better considered as a Short Report. In order to do this, you will need to reduce the number of main Figures to 4; please can you do this by either combining them or by moving some of them to the Supplement. I note that some of your Figures are very large and detailed, so if you're worried about this resulting in a loss of legibility, you could provide higher resolution versions of the main Figs in the Supplement. Please could you also select "Short Report" from the choice of article type? There is no need for any other formatting changes.

Please re-submit your manuscript within two working days, i.e. by Sep 11 2019 11:59PM.

Kind regards,

Roli Roberts

Senior Editor

PLOS Biology

---

## [Decision Letter · Decision Letter 1]

21 Oct 2019

Dear David,

Thank you very much for submitting your manuscript "Patterns of smallpox mortality in London, England, over three centuries" for consideration as a Short Report at PLOS Biology. Your manuscript has been evaluated by the PLOS Biology editors, an Academic Editor with relevant expertise, and by four independent reviewers.

You'll see that all four reviewers are broadly positive about the study. However, reviewer #2 (a historical demographer) raises some substantial concerns about the historical aspects, including record-keeping practices over the years; please note that her main review is in the Word attachment. In addition, reviewers #3 and #4 ask for you to expand further on your analyses. Please can you attend to all the concerns raised.

In addition, I think I might've been a bit hasty in asking you to compress the Figures to fit our "Short Report" format. Given the overall enthusiasm expressed by reviewers #1, #3 and #4, the Academic Editor and I would be happy for you to re-expand the Figures and text (as needed) and change back to a full Research Article, should you wish.

In light of the reviews (below), we will not be able to accept the current version of the manuscript, but we would welcome resubmission of a much-revised version that takes into account the reviewers' comments. We cannot make any decision about publication until we have seen the revised manuscript and your response to the reviewers' comments. Your revised manuscript is also likely to be sent for further evaluation by the reviewers.

Your revisions should address the specific points made by each reviewer. Please submit a file detailing your responses to the editorial requests and a point-by-point response to all of the reviewers' comments that indicates the changes you have made to the manuscript. In addition to a clean copy of the manuscript, please upload a 'track-changes' version of your manuscript that specifies the edits made. This should be uploaded as a "Related" file type. You should also cite any additional relevant literature that has been published since the original submission and mention any additional citations in your response. 

Before you revise your manuscript, please review the following PLOS policy and formatting requirements checklist PDF: http://journals.plos.org/plosbiology/s/file?id=9411/plos-biology-formatting-checklist.pdf. It is helpful if you format your revision according to our requirements - should your paper subsequently be accepted, this will save time at the acceptance stage.

Please note that as a condition of publication PLOS' data policy (http://journals.plos.org/plosbiology/s/data-availability) requires that you make available all data used to draw the conclusions arrived at in your manuscript. If you have not already done so, you must include any data used in your manuscript either in appropriate repositories, within the body of the manuscript, or as supporting information (N.B. this includes any numerical values that were used to generate graphs, histograms etc.). For an example see here: http://www.plosbiology.org/article/info%3Adoi%2F10.1371%2Fjournal.pbio.1001908#s5.

For manuscripts submitted on or after 1st July 2019, we require the original, uncropped and minimally adjusted images supporting all blot and gel results reported in an article's figures or Supporting Information files. We will require these files before a manuscript can be accepted so please prepare them now, if you have not already uploaded them. Please carefully read our guidelines for how to prepare and upload this data: https://journals.plos.org/plosbiology/s/figures#loc-blot-and-gel-reporting-requirements.

Upon resubmission, the editors will assess your revision and if the editors and Academic Editor feel that the revised manuscript remains appropriate for the journal, we will send the manuscript for re-review. We aim to consult the same Academic Editor and reviewers for revised manuscripts but may consult others if needed.

We expect to receive your revised manuscript within two months. Please email us (plosbiology@plos.org) to discuss this if you have any questions or concerns, or would like to request an extension. At this stage, your manuscript remains formally under active consideration at our journal; please notify us by email if you do not wish to submit a revision and instead wish to pursue publication elsewhere, so that we may end consideration of the manuscript at PLOS Biology.

When you are ready to submit a revised version of your manuscript, please go to https://www.editorialmanager.com/pbiology/ and log in as an Author. Click the link labelled 'Submissions Needing Revision' where you will find your submission record. 

Sincerely,

Roli

Senior Editor

PLOS Biology

REVIEWERS' COMMENTS:

Reviewer #1:

[identifies himself as David N. Fisman]

This is a magnificent paper, and highly informative. It is mathematically sophisticated but clearly written and easily understood by non-mathematicians.

I have 2 minor comments:

1. Unless I'm mistaken, "prodrom" is not a variant spelling of "prodrome". At any rate, prodrome (with e) is the more familiar term.

2. Eczema vaccinatum isn't "severe eczema". It represents extensive cutaneous infection in individuals with pre-existing eczema or atopic dermatitis. This can happen with other viruses too (c.f., eczema herpeticum). Would revise terminology.

Reviewer #2:

[identifies herself as Romola Davenport]

IMPORTANT: More detailed comments from this reviewer are available in the attached, downloadable Word document.

This paper reports for the first time an analysis of high frequency counts of smallpox burials and deaths in London over almost 300 years. The authors juxtapose their analyses with historical factors, especially relating to the smallpox control measures, that they claim account for the major changes they identify in smallpox mortality matterns in London. The patterns presented are very interesting however the paper's conclusions are not justified by the method used. The method adopted relies on visual comparison of smallpox patterns with an historical timeline. This is unsatisfactory, but would suffice for an exploratory paper that was designed to point the way to further research. However at present the method is used with insufficient rigour with respect to data quality and historical accuracy. 

I recommend that the authors make further adjustments to the smallpox burial data to correct for known shortcomings, and rewrite sections of the paper to enhance clarity and to acknowledge more fully the tenuous nature of some of the claims regarding the effects of smallpox control measures. These recommendations are set out more fully in an attached document.

Reviewer #3:

This article reports the first analyses of a deeply fascinating dataset: the records of smallpox mortality in London from 1664-1930. Digitizing and making publicly available this dataset is a tremendous advance in and of itself, as there are a large number of follow-up studies that will be able to explore different features of this dataset and use it to uncover the influences of demographic, social, and public health change on the dynamics of infectious disease. It is therefore an important contribution, worth publication in a journal with the exposure of PLoS Biology.

As a Short Report, I realize that the analyses do not necessarily need to be complete. And yet, I was left wanting more from even these preliminary analyses. In particular, the article goes into extensive detail about the history of smallpox in London, but the connection between the statistical analyses and this story is only explored in a very limited fashion. In particular, while Fig. 2 and Fig. 4 both annotate the data with major historical events that might be relevant to smallpox transmission and mortality, the discussion of the connection between any patterns observed in the data and these events in the Results or Discussion is very limited. For example, on lines 454-462, the authors note several timepoints where seasonality shifts, but those timepoints are not explicitly related back to any of the key events in the history of smallpox. Or, for another example, on lines 422-428, there is more connection between the regularity of epidemics and the history of variolation and vaccination, but there is no discussion of whether the patterns observed in the data (e.g. the shift to very regular epidemics with the introduction of variolation) make sense given the likely effects of control measures on transmission. This carries over into the Discussion as well, which was too brief given all of the potential for a very interesting discussion of a very interesting dataset.

I do think there is plenty of space available to expound on the connection between the statistical analyses and the historical transitions in both London's demography and the control of smallpox, as the historical sections can probably be shortened or eliminated without much loss. In particular, the entire section on "Types of Smallpox" (including Fig. 1) could be removed without loss of content, as none of the information in that section has any bearing on the analysis presented in the paper. Much of that information (e.g., on incubation periods and infectiousness) would be relevant to an paper attempting to fit an epidemiological model to the data, which I assume will happen in a follow-up paper. Similarly, the section, "Smallpox history" could also be drastically shortened: it is included only "to provide context" (line 27), but it is unclear to me how that context is actually helpful, given the detailed discussion of smallpox history in London that follows, especially in the "Annotation of data with historical events" section.

In sum, I think this is a very interesting dataset and analysis, and that some effort to tighten the connection between that analysis and the historical events discussed in the paper would make it a very valuable contribution.

Reviewer #4:

This is a unique and fascinating study on the long-term population dynamics of smallpox. The assembling and curating of such a long record with concurrent relevant events on the history of control is of major value to the field of population dynamics of infectious diseases in general.

I have however a number of comments intended to clarify the analyses and to strengthen the conclusions. I believe the authors should be able to address these, and to go further in the interpretation of the results and the kinds of questions this data set, unique in its length, will allow them or others to address in the future, given the results here and what is already known about seasonal SIR dynamics. On this last aspect, I have found the paper a bit thin. 

1) The changes in seasonality are an important result. A main conclusion is a shift from summer to winter (line 490 Discussion). The analysis presented in Fig 4b is difficult to ‘see’. That is, one can read the caption and text but it is difficult to actually see the described patterns in the figure itself. This is because with so many years compressed in the x axis, one cannot easily follow where the main season is for a given year and how the trends go. I would recommend additional plots to make this sufficiently clear. For example, boxplots representative of the seasonal patterns in selected windows of time would help despite the non-stationarity of the ‘average’ seasonal cycle. (A smaller comment: the sentence on the zeros not being represented in the caption was confusing. The effect of the detrending needs to be clarified). Another important conclusion on the seasonality is in Line 471, where the authors write about changes in the strength of the annual power. I could not see where this comes from. The Fourier power spectrum averages over time, and as far as I could tell, the wavelet spectrum, which doesn’t, was applied after filtering the annual variation, so one cannot see there the non-stationarity of the annual power. 

2) Another major result concerns the changes over time of the interannual variability; that is, the changes in the dominant periods of multi-annual cycles. These changes are interpreted as the consequence of control rather than birth rates (and contrasted to the importance assigned to demography in studies of other seasonal infectious diseases). This is where I would have liked to see a clearer exposition of why demography is not an important driver of the changes. From previous influential work on measles and childhood infections (including contributions of one of the authors, David Earn), we know that birth rates, transmission rates, and vaccination coverage, can have equivalent/related effects on seasonal SIR dynamics. How do the patterns/trends here relate to that previous knowledge and how can one infer the importance or dominance of control over demography?

3) Are there ways to look at the changes in the intensity of the annual cycle in relation to the importance of other, longer, cycles? We know from earlier theory how the spectrum of seasonal SIR dynamics should change with demography and transmission intensity. Are there concurrent patterns in how the power is distributed over different periods that would relate to that theory and allow interpretation?

4) Similarly, a major contribution of this paper will be to make this remarkable data set available to the community. The analyses in the paper itself are largely descriptive of trends in the seasonal and interannual variability and in relation to the timing of control efforts. It would be valuable to go further and based on these patterns, provide some major directions for what these data will allow going forward, in particular from their analyses with process-based models and related hypothesis about smallpox and seasonal SIR dynamics in general.

---

## [Decision Letter · Decision Letter 2]

10 Aug 2020

Dear Dr Earn,

Thank you for submitting your revised Research Article entitled "Patterns of smallpox mortality in London, England, over three centuries" for publication in PLOS Biology. I have now obtained advice from three of the original reviewers and have discussed their comments with the Academic Editor. 

Based on the reviews, we will probably accept this manuscript for publication, assuming that you will modify the manuscript to address the remaining points raised by the reviewers. IMPORTANT: The article type still seems to be "Short Report"; please change it to "Research Article" when re-submitting. Please also make sure to address the Data Policy-related requests noted at the end of this email.

We expect to receive your revised manuscript within two weeks. Your revisions should address the specific points made by each reviewer. In addition to the remaining revisions and before we will be able to formally accept your manuscript and consider it "in press", we also need to ensure that your article conforms to our guidelines. A member of our team will be in touch shortly with a set of requests. As we can't proceed until these requirements are met, your swift response will help prevent delays to publication.

*Copyediting*

*Published Peer Review History*

*Early Version*

*Submitting Your Revision*

Sincerely,

Roli Roberts

Senior Editor,

rroberts@plos.org,

PLOS Biology

DATA POLICY:

We note that your raw data are deposited in http://iidda.mcmaster.ca - however, we strongly prefer more stable, non-institutional repositories (e.g. Dryad, Figshare, Github), and request that you make such provision for depositing your data and code. At the moment http://iidda.mcmaster.ca is giving a timeout error, which gives us further for the long-term availability of this important dataset.

In addition, we ask that all individual quantitative observations that underlie the data summarized in the figures and results of your paper be made available in one of the following forms:

Regardless of the method selected, please ensure that you provide the individual numerical values that underlie the summary data displayed in the following figure panels as they are essential for readers to assess your analysis and to reproduce it: Figs 1, 2, 3, 4, 5, 6, S1, S2. NOTE: the numerical data provided should include all replicates AND the way in which the plotted mean and errors were derived (it should not present only the mean/average values).

REVIEWERS' COMMENTS:

Reviewer #2:

[identifies herself as Romola Davenport]

The authors have done a great job in revising the paper. It is much clearer and more tightly written, and presents a very impressive integration of historical and epidemiological literatures. I think it will make a great addition to current debates over the recent evolution of smallpox. 

I have only minor comments that should be addressed before publication. 

1. The authors are now perhaps too reticent in attributing causation, especially with respect to vaccination (page 14/33, lines 431-6). The decline in smallpox deaths with the introduction of vaccination in the early nineteenth century is very marked in both raw and normalised burials. This phenomenon was observed in other cities and states that adopted vaccination, and coincided with a marked decline in all-cause mortality (so the reduction in normalised smallpox burials is likely to underestimate the fall in smallpox mortality). 

2. The term 'mortality' usually refers to mortality *rates*, that is, deaths per population at risk. The authors should distinguish clearly when they are talking about counts of deaths or normalised deaths, to avoid confusion. Figure 1 top panel should be labelled as smallpox deaths, not mortality, and the y-axis should read 'weekly smallpox deaths'. The y-axis of Figure 2 should also be labelled 'weekly all-cause deaths'.

3. A slightly larger comment: Why were normalised deaths used to study seasonal patterns? Seasonal patterns in raw deaths should be largely unaffected by longer-term changes in reporting units or under-registration (for the same reasons that normalised deaths are preferable for other purposes). The use of raw deaths to study seasonality would avoid the potential distortions caused by other seasonal patterns of mortality. For example, scarlet fever emerged as a major cause of death in London in the 1930s, with a marked autumnal pattern. This could have reduced the proportion of all deaths due to smallpox in the autumn, regardless of the underlying seasonal pattern of smallpox mortality in this period. Other important causes of death also showed seasonal patterns, and some of these changed over the period of the study (including measles). The authors should acknowledge this potential problem, if they prefer to use normalised burials and deaths. 

4. Table 1 (appendix B): the labels for the third and fourth columns appear to be transposed. 

5. page 3/33 line 51: another important element in the eradication of smallpox was the relatively low infectivity of smallpox. 

6. page 3/33 line 60: insert 'and only for a few towns' between 'until later' and 'Bills of Mortality'. 

7. page 4/33, line 81: perhaps replace 'exists' with 'survives' (to avoid the impression that patchy series necessarily imply gaps in the production of weekly bills as opposed to survival).

8. page10/33, line 256-7: perhaps add 'need for periodic revaccination' to the list of impediments to vaccine uptake. There was some resurgence of smallpox, and a rise in average age of victims, in the 1820s and 1830s that may have been associated with the waning of vaccine-derived immunity in birth cohorts in which vaccination was very common. 

9. Typographical errors: abstract line 1, 'devastated' for 'devasted'; page 14/33, line 427: 'and' for 'an'; page 14/35, line 434: insert 'of' after 'introduction'.

Reviewer #3:

I appreciate the revisions that went into this manuscript, and I think it is very close to publication-ready at PLoS Biology. I remain convinced that the data, by itself, is incredibly valuable, and the extended analysis presented here makes this paper a more meaningful contribution as well. I have only a few comments that should be straightforward to address.

I feel that the importance (the "so what" question) of the analyses presented here needs to be better articulated in the Introduction and Discussion sections. Right now the value is implicit throughout the manuscript, and nowhere do the authors state clearly what they can learn from the statistical analysis of this data, and how will what they learn from these analyses be useful, both for understanding infectious diseases more generally and also for the next phase of work on smallpox specifically (e.g., building mechanistic models).

On the description of the handling of the heaped data, you might note that you considered other ways of handling the heaped data (e.g., the way it was handled in the first version of the manuscript) and that this did not have any effect on the conclusions drawn here.

The rationale for identifying the "Intervention uptake levels" in Fig. 1 is never made clear. Why, for example, does the assumed uptake level go from "very low" to "low" in 1728? Why does it go from "low" to "moderate" in 1740, if the first charitable variolation hospital didn't open until 1746? Etc. I realize that this doesn't impact the analyses presented here (because you are not seeking to draw any quantitative conclusions between the dynamics and the level of intervention uptake), but I still think it would be useful to provide some justification.

You do not justify the use of square-root transformation in the spectral analysis section. Also, in this section it would be useful to explain to a reader who has limited exposure to time series analysis what is gained by carrying out both the power spectrum analysis and the wavelet analysis. 

You are missing an "of" on line 434 between "introduction" and "vaccination."

There is some inconsistency in how Fig. 4B is discussed in the Results and Discussion. In the Results, the power of the annual period is not discussed - you only mention periods at 2, 3, and 5 years. But in the Discussion (lines 465-466), you say that "the wavelet spectrum in Fig. 4B shows a peak at one year," a finding that is not very apparent in Fig. 4B (at least to me).

The discussion of possible viral evolution (lines 551-559) could reference some of the theoretical work by Sylvain Gandon and Troy Day on how vaccination is expected to drive pathogen evolution (especially the reference on line 557-559 that suggests the potential for variolation to be "leaky"). E.g., 

Gandon, S., Mackinnon, M. J., Nee, S., & Read, A. F. (2001). Imperfect vaccines and the evolution of pathogen virulence. Nature, 414(6865), 751-756.

Gandon, S., & Day, T. (2007). The evolutionary epidemiology of vaccination. Journal of the Royal Society Interface, 4(16), 803-817.

To help foreshadow the future work you anticipate in response to these data and analyses, you might say a bit more about *how* the studies of measles and other childhood infections were able to explain dynamical transitions evident in the data (liens 583-588).

You are missing an "in" on line 610 between "patterns" and "infectious."

Reviewer #4: 

The authors have done a great job at clarifying my questions and addressing my concerns in the revision. The revised presentation on the changes in seasonality and on the interannual variation is now very clear. I also appreciated the discussion on the directions these data and patterns open up for future research.

---

## [Editor Report · Decision Letter 3]

17 Nov 2020

Dear Dr Earn,

On behalf of my colleagues and the Academic Editor, Andy P. Dobson, I am pleased to inform you that we will be delighted to publish your Research Article in PLOS Biology. 

PRODUCTION PROCESS

Before publication you will see the copyedited word document (within 5 business days) and a PDF proof shortly after that. The copyeditor will be in touch shortly before sending you the copyedited Word document. We will make some revisions at copyediting stage to conform to our general style, and for clarification. When you receive this version you should check and revise it very carefully, including figures, tables, references, and supporting information, because corrections at the next stage (proofs) will be strictly limited to (1) errors in author names or affiliations, (2) errors of scientific fact that would cause misunderstandings to readers, and (3) printer's (introduced) errors. Please return the copyedited file within 2 business days in order to ensure timely delivery of the PDF proof. 

If you are likely to be away when either this document or the proof is sent, please ensure we have contact information of a second person, as we will need you to respond quickly at each point. Given the disruptions resulting from the ongoing COVID-19 pandemic, there may be delays in the production process. We apologise in advance for any inconvenience caused and will do our best to minimize impact as far as possible.

EARLY VERSION

PRESS 

Kind regards,

Vita Usova

Publication Assistant, 

PLOS Biology

on behalf of

Roland Roberts,

Senior Editor

PLOS Biology